# Burnout Syndrome and Sleep Quality in Basic Education Teachers in Mexico

**DOI:** 10.3390/ijerph20136276

**Published:** 2023-07-01

**Authors:** Francisco Sánchez-Narváez, Juan Jesús Velasco-Orozco, Eduardo Pérez-Archundia

**Affiliations:** 1Faculty of Humanities-Enterprise, Universidad Estatal del Valle de Ecatepec, Valle de Anahuac, Ecatepec 55210, Mexico; 2Mexican Institute of Integral Sleep Medicine, del Valle, Benito Juárez 03100, Mexico; 3Faculty of Anthropology, Universidad Autónoma del Estado de México, Universidad, St. Toluca 50130, Mexico; 4Instituto Superior de Ciencias de la Educación del Estado de México, Santa Cruz, Toluca 50030, Mexico

**Keywords:** burnout syndrome, sleep quality, depression, anxiety

## Abstract

Burnout syndrome (BS) is the result of chronic stress in the workplace. Moreover, chronic stress can affect sleep. A unidirectional relationship has been established between burnout and sleep, and it is known that white-collar workers with burnout syndrome have sleep fragmentation and marked daytime sleepiness. Objective: The aim of this study was to assess the relationships between burnout and sleep quality in elementary school teachers in Mexico. Methods: We collected data from more than 400 teachers who completed tests. Correlation analyses controlled for anxiety and depression, and Poisson logistic regression analyses were performed to examine the relationships of burnout with sleep quality, depression, and anxiety. Results: There was a significant correlation between burnout syndrome (mainly in the dimension of emotional exhaustion) and sleep disturbances; significant correlations were also observed with other burnout, depression, and anxiety dimensions. The strength of the correlations decreased after controlling for depression and anxiety. Conclusions: The symptoms of burnout syndrome in teachers can overlap with sleep disorders, so it is necessary to make a differential diagnosis to differentiate burnout syndrome from depression and anxiety, among others.

## 1. Introduction

In Latin America, as elsewhere in the world, economic changes, social transformations, and the extension of longevity of the population have modified the health of the general population, with an increase in psychosocial problems such as burnout syndrome and mental illness. Epidemiological studies conducted in the last decade have indicated that there is an increase in these disorders. The mental health of teachers plays a very important role in the quality of education and in the processes related to teaching and learning, affecting not only teachers but also students [1].

These disorders also influence other aspects of education, such as teacher attendance and participation, which are important because they affect the academic, organizational, and administrative structure of the institution indirectly. These illnesses generate a high rate of absenteeism, work incapacity, and premature retirement. One of the disorders that significantly affects teachers is burnout syndrome, which is a response to chronic stress.

Burnout has been described as a syndrome of exhaustion and indifference that occurs in workers who offer services or provide help to people, which has the following three dimensions: (a) emotional exhaustion: the person feels that they can no longer give more of themselves in the affective or psychological sphere; (b) depersonalization: the development of negative attitudes and insensitivity towards colleagues, clients or recipients of services, associated with a somewhat cynical and impersonal attitude and isolation from others; and (c) diminished self-fulfilment: having a perception that the possibilities of achievement at work have disappeared, coupled with experiences of failure and feelings of low self-esteem [2,3]. Reports in the literature worldwide consider it one of the main syndromes affecting workers’ health, with teaching professions accounting for 27 percent of burnout cases; it is second in prevalence only to health service professions [4]. Globally and in Mexico, these data for the working population are controversial because different instruments have been used for assessments, and studies focusing on teachers are scarce [5].

### 1.1. Burnout Syndrome, Depression, and Anxiety

Few studies that have evaluated the most prevalent mental illnesses in the population, particularly for specific occupations, such as teaching. Depression is characterized by anhedonia, i.e., the loss of interest or pleasure, depressed mood, fatigue or loss of energy, impaired concentration, and feelings of worthlessness, decreased or increased appetite, sleep problems (hypersomnia or insomnia), and suicidal ideation. Anxiety serves as a protective mechanism when faced with potentially hazardous situations. Physical symptoms associated with anxiety disorders may include sleep disturbances, muscle tension, and feelings of restlessness. In Mexico, the prevalence of depression in the population is estimated at 6.2 to 8.1 percent, and that of generalized anxiety disorder is estimated at 5.5 percent [6]. However, in other countries, a higher prevalence of depression and anxiety has been found in the teacher population. Among teachers in England, 23 percent have depression and 37 percent have anxiety [7]. In addition, 19.4 percent of teachers have moderate to severe depressive symptoms [8]. In Italy, it has been reported that over 49 percent of teachers have depression and 11 percent have anxiety [9]. In Hong Kong, the prevalence of anxiety is estimated at 75.8 percent [10], and in France, the prevalence of mental disorders (phobias, panic disorders or obsessive compulsive disorders) in female teachers is reported to be 16.4 percent higher than in male teachers (7.7 percent), whereas mood disorders are reported to occur in 44.6 percent and 25.6 percent of female and male teachers, respectively [11].

Studies in Mexico on BS in teachers have mainly focused on studying psychosocial factors, including stress, as its cause [12,13,14]. With regard to other occupational groups at risk, such as doctors and nurses, there have been some studies that address the analysis of BS, depression, and anxiety [15,16,17]. These studies independently identified the presence of the disorders but did not assess the interactions between them. Additionally, these publications addressed the study of burnout, as well as depression and anxiety, with different instruments and concepts, so the ability to conduct comparisons between these data are significantly limited [5].

Although a rigorous delimitation is proposed [18], it is difficult to clearly define the boundaries between Burnout syndrome, depression, and anxiety. There are many clinical similarities between these disorders; we can observe fatigue, irritability, decreased performance, and feelings of distress both at work and in interpersonal relationships. Other symptoms include absenteeism, decreased physical activity, and difficulty concentrating or remembering [19,20]. Some authors refer to burnout as a subtype of depression rather than a separate disorder [21].

According to several studies [22,23,24], there is a relationship between burnout and the presence of depression and anxiety. It has also been documented that people with burnout have a twelve times higher risk of depression and up to a thirteen times higher risk of anxiety [25]. This suggests a significant association between these disorders.

However, the relationship and difference between burnout syndrome and these diseases has been limited by its definition and criteria for classification; empirical research remains insufficient and contradictory in some cases [26].

### 1.2. Burnout Syndrome and Sleep Quality

Several studies have established a relationship between burnout syndrome and sleep quality in various occupational populations, including doctors, nurses, social workers, and teachers [27,28,29,30,31].

The proportion of teachers with poor sleep quality ranges from 55 percent to 61 percent [32,33]. This percentage increased by approximately 74% during the COVID-19 pandemic. Additionally, an increase in symptoms of depression and anxiety was observed during this time period [34,35]. The consequences of poor sleep quality have been associated with several factors. In one study, it was found that teachers with inadequate sleep quality (38%) experienced failure at work, social exclusion, and emotional dissonance more frequently during working their hours than teachers with adequate sleep quality [36]. In relation to sex, it has been reported that female teachers with inadequate sleep quality go to bed later and get up earlier than men due to the additional workload of domestic chores and professional demands, resulting in less time in bed, more daytime sleepiness, and poorer sleep quality [30]. In a review that included teachers who slept less than 6 h a day, it was observed that teachers have an increased occurrence of voice disturbances [37]. Teachers with poor sleep quality had higher body mass index (BMI) values and were approximately twice as likely to report chest pain, wrist pain, back pain, and ankle/foot pain [38].

Additionally, sleep quality has been associated with mental health issues, burnout syndrome, depression, and anxiety. Söderström et al., 2012 [29] found a significant relationship between burnout syndrome and poor sleep quality, with workers who slept less than 6 h being more affected. Significant relationships have been reported between burnout and emotional exhaustion, depersonalization, and personal accomplishment. Linear regression analyses have indicated that, as sleep quality scores increase (>5), the mean scores of emotional exhaustion and depersonalization show an upwards trend, while the mean score of personal accomplishment shows a downwards trend [39]. In this regard, significant positive correlations have been established between sleep quality, emotional exhaustion, and depersonalization, and negative correlations with respect to personal accomplishment. There is a positive correlation between emotional exhaustion and sleep quality, and sleep duration and daytime dysfunction; meanwhile, depersonalization and personal accomplishment have only been associated with daytime dysfunction [40,41].

Some research has established that the relationship between sleep quality, depression, and anxiety occurs in a bidirectional manner. In previous studies, some physiological parameters associated with sleep and depression have been evaluated; a decrease in rapid eye movement (REM) latency (REM latency ¼ interval between sleep onset and the onset of the first REM sleep period), an increase in total REM sleep time and REM density (i.e., the frequency of rapid eye movements per REM period), and a decrease in the production of slow wave sleep (SWS) were observed. These parameters can be used to predict the presence of depression [42]. In subjective measurements between depression and sleep quality, a significant correlation was observed with respect to all aspects of sleep quality (sleep latency, daytime dysfunction, sleep disturbances, among others). Yu et al., 2016 [43] found that symptoms of depression and anxiety were highly associated with greater sleep disturbances. In addition, anxiety symptoms were associated with higher sleep latency and lower sleep quality, while depression symptoms were linked to higher levels of daytime dysfunction.

It Is important to note that between 50 percent and 80 percent of the population with a mental disorder has a sleep disturbance, which is a symptom of 19 Axis I disorders [44]. From a clinical perspective, sleep disturbances can be considered an epiphenomenon that will disappear once the primary mental disorder is treated and not as a valid, independent, clinical entity [44,45].

One of the important problems regarding burnout syndrome, its identification, assessment, and diagnosis is that it is often confused with other disorders such as sleep disturbances, depression, and anxiety. The aim of this study is to describe the levels of burnout and sleep quality in elementary school teachers. It also analyses the relationship between these variables when depression and anxiety are taken into account. Our hypothesis is that poor sleep quality is related to higher burnout (higher emotional exhaustion, lower depersonalization, and lower personal accomplishment). Additionally, the symptoms that occur between these conditions can overlap and even coexist simultaneously, and they may also present as independent entities.

## 2. Materials and Methods

### 2.1. Survey Design

The teachers selected to participate in this study were from a nonrandomized sample and worked at the primary and secondary school levels in public schools affiliated with the Subsecretaría de Educación Básica y Normal del Estado de México in Mexico. A total of 1211 teachers were selected, including 616 primary education teachers and 515 secondary education teachers. The number of teachers calculated for the sample size was 476, accounting for 39.3 percent of the total population (1211), with a precision of 3.5%. A total of 427 teachers were included in the sampled population, including 251 primary education teachers and 176 secondary education teachers, which was 10.3 percent lower than the calculated sample size.

### 2.2. Procedure

The sociodemographic questionnaire and the tests were administered collectively and self-administered to the teachers, respectively. The estimated time for giving directions, answering the questionnaire, and completing the tests was 50 min on average.

We requested a meeting with the supervisors at least one week prior to administering the questionnaires to obtain their permission. To facilitate the application of the tests, the day of the technical council meeting was chosen. The technical council meets on the last Friday of each month. This type of meeting takes place in the school centre at a time and place determined by the supervisor. The meetings are attended by all teachers belonging to the school zone in their respective working hours. In two primary school supervisions, due to organizational reasons, another appointment was scheduled for the administration of the questionnaires. During this visit, data were collected for 50 teachers from the morning shift and 41 teachers from the afternoon shift.

In addition, the teachers were provided with the following information:—A brief explanation of the purpose of the research was given.

-Their consent to participate in the research was requested.-Emphasis was placed on confidentiality and use of the data obtained.

Due to the noninvasive nature of this research, the recommendations of the agreement were followed. The institution does not have a bioethics committee. This study followed strict bioethical principles in accordance with the General Law on Health in Research (DOF 02-04/2014) and the Law on the Protection of Personal Data Held by Private Parties (DOF 05-07/2010) in Mexico. The anonymous database ensured that the privacy of study participants was always maintained.

### 2.3. Instruments

#### 2.3.1. Maslach Burnout Inventory

The Maslach Burnout Inventory for Education (MBI-Ed) [46] was designed to assess BS according to three fundamental aspects. The test is composed of 22 items and consists of three dimensions divided into the following categories: (a) emotional exhaustion (Cronbach’s alpha 0.90), (b) depersonalization (Cronbach’s alpha 0.76), and (c) personal accomplishment (Cronbach’s alpha 0.76). To evaluate and interpret the results, the scores for each dimension were categorised as low, medium, and high [3]. For the distribution of the values, the 33rd and 66th percentiles of the total scores per dimension were considered. The criteria for designating a case of burnout according to Maslach and Jackson 1981 [2] are as follows:High level of burnout: a high level of burnout is considered when the score of the two dimensions of emotional exhaustion (AE) and depersonalization (DP) are high and that of personal accomplishment (PA) is low.Average level of burnout: the average result of the three subscales is medium.Low level of burnout: the results of the emotional exhaustion (AE) and depersonalization scales (DP) are low, and the score in the dimension of personal accomplishment (PA) is high.The estimated scores to classify the dimensions in this sample were determined from the 33rd and 66th percentiles.In this context, we can distinguish burnout cases from cases that do not meet these criteria; for example, a case that scores high on EE, PD, and PR is not considered burnout. Based on this criterion, the variable was dichotomized to compare burnout cases (severe, moderate, and mild) with non-burnout cases. Percentage prevalence rates were calculated based on this criterion.

#### 2.3.2. The Pittsburgh Sleep Quality Inventory (PSQI)

The self-administered questionnaire consists of 24 questions; however, only the answers to the first 19 questions are used to obtain the overall score. The 19 items are grouped into seven components with each scored on a scale from 0 to 3. For example, for a sleep disorder, 0 represents no sleep disorder, 1 represents mild sleep disturbance, 2 represents moderate sleep disturbance, and 3 represents severe sleep disturbance. The sum of these seven components yields an overall subjective sleep quality score (range 0–21); higher scores indicate poor subjective sleep quality, and a PSQI ≥of 5 is associated with poor sleep quality [47].

Using this criterion, we can dichotomize the cases that have a score >5 to calculate the percent prevalence for overall sleep quality.

For the Poisson analysis, the remaining seven categories were dichotomised. For example, the very good and fairly good components were considered good sleep quality, while the fairly bad and very bad components were considered poor sleep quality. It was considered adequate sleep latency when LS was less than 15 min, adequate sleep duration when SLD was greater than 7 h, and adequate sleep efficiency when SE was greater than 85. Dichotomization was performed for the SD, USM, and DD categories as follows: The first group was formed from the categories “not during the last month” and “less than once a week”. The second group was formed from the categories “once or twice a week” and “three or more times a week”. The dichotomous reference categories used in the Poisson analysis were those related to sleep dysfunction or disturbance.

The questionnaire has a high coefficient of internal homogeneity (Cronbach’s alpha = 0.83) [47]. The version of the PSQI validated in Mexico has a Cronbach’s alpha of 0.75 [48].

#### 2.3.3. The Beck Depression Inventory

The Beck Depression Inventory (BDI) was developed in 1961 to assess the severity of depressive symptoms. It was translated and standardized into Spanish in 1998. It is a self-administered test with a Cronbach’s alpha of 0.87 [49], which consists of 21 items, from which responses are graded on a Likert-type numerical scale, where scores range from 0 to 3. It is scored according to the criteria established in the Mexican version of the BDI on a scale of 0–63 points, with each response having a value of 0–3 points. In this study, the total score was obtained and interpreted as follows: ≤9, no depressive symptoms; 10–16, mild symptomatology; 17–29, moderate; and ≥30 points, severe depressive symptomatology.

#### 2.3.4. The Beck Anxiety Inventory

The Beck Anxiety Inventory (BAI) was developed in 1988 to assess the severity of anxiety symptoms with a Cronbach’s alpha of 0.87 [50]. The Beck Anxiety Inventory was translated into Spanish. It is scored using a 0–63 point scale, with each response worth 0–3 points. The total score can be interpreted as follows: ≤7, no anxiety symptomatology; 8–15, mild symptomatology; 16–25, moderate; and ≥26, severe anxiety.

Anxiety and depression, along with their respective classification levels, were used to distinguish individuals with the disorder from those without. Poisson regression analysis was conducted using this dichotomization.

### 2.4. Statistical Analysis

-SPSS version 25 was used to carry out descriptive and analytical statistical analyses.-Regarding the assessments of sociodemographic and occupational variables, continuous variables are reported as the mean and standard deviation, and categorical variables are reported as percentages. Descriptive statistics were used for classification and analysis, with frequencies and percentages reported.-Sleep quality characteristics are presented in terms of frequencies and percentages for each variable.-Regarding the assessments of emotional exhaustion, depersonalization, and lack of personal accomplishment, mean scores and standard deviations for these three dimensions of burnout syndrome are presented. Individuals with burnout syndrome were categorized according to severity (mild, moderate, or severe) as well as sleep quality and symptoms of depression and anxiety.-In the analysis of burnout syndrome dimensions, the Mann–Whitney U test was used to compare continuous variables between two groups, and the Kruskal–Wallis test was used for comparisons among three or more groups. Means and standard deviations were determined to detect significant group differences in the burnout dimensions and sociodemographic and occupational variables. A value of *p* ≤ 0.05 was considered significant.-Spearman correlation analysis was used to examine the relationships of burnout syndrome with sleep quality, anxiety, and depression. Poisson logistic regression analysis (odds ratio) was used to identify variables predictive of burnout. Other regression analyses were not performed because the dimensions of burnout, sleep disturbances, anxiety, and depression did not meet the assumptions of normality of distribution, equality of variance, or a linear relationship.

## 3. Results

### 3.1. Sociodemographic Data

The sample comprised 427 teachers, of whom 276 (60 percent) were female and 171 (40 percent) were male. The sociodemographic characteristics indicated that most of the teachers were aged between 36 and 46 years (36.3 percent) with an average age of 40.8 years (SD +/− 9.1). The lowest age was 22 years, and the highest age was 64 years.

Among these teachers, 66% were married, 81% had one or more children, 41.7% worked 40 h or more per week, and 53% worked between 20 and 39 h per week. Additionally, 72% of teachers had 31 or more students under their care, and approximately 54% had two jobs. The majority of teachers (92%) were permanently employed.

There are significant differences in sleep quality between the groups for three variables (SL, USM and DD), which are associated with factors such as age, marital status, educational level, and number of students (Table 1).

### 3.2. Sociodemographic Data and Burnout Syndrome

#### 3.2.1. Emotional Exhaustion

According to socio-demographic variables, there was a significant difference in emotional exhaustion depending on the number of hours spent in the workplace. Those working between 5 and 15 h per week were less exhausted than those working more than 40 h per week ($\tilde{x}$ = 11.9 SD ± 5.4 and $\tilde{x}$ = 17.5 SD ± 9.3).

#### 3.2.2. Depersonalization

There were significant differences in depersonalization with respect to sex. Depersonalization was greater in men than in women ($\tilde{x}$ = 5.1 SD ± 4.5 and $\tilde{x}$ = 4.2 SD ± 4.3), respectively). Additionally, differences were observed between teachers with more than three children ($\tilde{x}$ = 5.3 SD ± 4.3) compared to those with one to two children ($\tilde{x}$ = 4.2 SD ± 4.4).

#### 3.2.3. Personal Accomplishment

On the personal accomplishment scale, there was a significant difference between the group aged 47 to 64 years ($\tilde{x}$ = 37.5 SD ± 7.8) and the group aged 25 to 35 years ($\tilde{x}$ = 35.4 SD ± 7.3). No statistically significant differences were observed between other sociodemographic variables and the dimensions of burnout syndrome (Table 2).

### 3.3. Sleep Quality

Table 3 shows the quality of sleep among the teaching population. Approximately 78 percent of teachers had a sleep latency of less than 30 min; 22 percent of the teachers had a sleep latency of more than 31 min, 25 percent of the teachers slept more than 7 h and 38.2 percent slept less than 6 h. The sleep efficiency of 64% of teachers was normal. For sleep disturbances, only 5 percent of the teachers reported sleep disturbances “Not during the past month”, while 69 percent indicated having disturbances “Less than once a week”. Approximately 93 percent of teachers reported not using any sleep medication and approximately 43 percent reported no daytime dysfunction. However, 21 percent of the teachers reported having daytime dysfunction which they indicated was a “problem” or a “big problem”. The mean PSQI score was $\tilde{x}$ = 6.23 SD ± 3.03. Overall, the quality of sleep in teachers was poor, and approximately 53.4 percent had some sleep disturbances.

### 3.4. Burnout Syndrome and Sleep Quality

Table 4 shows the means with their standard deviation corresponding to the dimensions of burnout and the various categories related to sleep quality.

#### 3.4.1. Emotional Exhaustion

Sleep quality: Significant differences in emotional exhaustion were observed when comparing individuals in the “Very bad” category of sleep quality with those in the “Fairly bad”, “Fairly good”, and “Very good” sleep quality categories ($\tilde{x}$ = 10.9 SD ± 7.9 vs. $\tilde{x}$ = 20.3 SD ± 8.9, $\tilde{x}$ = 14.9 SD ± 8.9, $\tilde{x}$ = 19.9 SD ± 11.6, respectively). There were also differences in emotional exhaustion between individuals with “Fairly good” sleep quality compared to those with “Fairly bad” sleep quality ($\tilde{x}$ = 14.9 ± 8.9 vs. $\tilde{x}$ = 20.3 ± 8.9).

Sleep latency: When comparing the time of subjective sleep onset <15 min with onset times of 16–30 min, 31–60 min, and >60 min, significant differences were observed in emotional exhaustion. Differences in emotional exhaustion were also observed when comparing the 16–30 min group with the >60 min group.

Sleep duration: With regard to sleep duration, there were statistically significant differences in emotional exhaustion between teachers who indicated sleeping more than seven hours and those who reported sleeping <5 and 5–6 h ($\tilde{x}$ = 13.4 SD ± 8.0 vs. $\tilde{x}$ = 19.2 SD ± 9.7 and $\tilde{x}$ = 17.7 SD ± 10.7). In addition, significant differences were found between the group that averaged 6–7 h of sleep compared to the group that averaged less than 5 h ($\tilde{x}$ = 15.6 SD ± 8.9 vs. $\tilde{x}$ = 19.2 SD ± 9.7).

Sleep efficiency: There were significant differences in emotional exhaustion between teachers who reported having at least 85 percent sleep efficiency and those who reported having 65–74 percent and less than 65% sleep efficiency ($\tilde{x}$ = 15.0 SD ± 9.0 vs. $\tilde{x}$ = 20.5 SD ± 9.4 and $\tilde{x}$ = 21.6 SD ± 11.9). There were also differences in emotional exhaustion between teachers who reported a sleep efficiency of 65–74 percent and those who reported a sleep efficiency of 75–84% ($\tilde{x}$ = 20.5 SD ± 9.4 vs. $\tilde{x}$ = 16.0 SD ± 9.2).

Sleep disturbances: Teachers who reported they had experienced sleep disturbances “Not during the past month” had significant differences in emotional exhaustion compared to the teachers who indicated they had sleep disturbances “Less than once a week” and “Three or more times a week” ($\tilde{x}$ = 11.5 SD ± 9.8 vs. $\tilde{x}$ = 14.4 SD ± 8.7 and $\tilde{x}$ = 30.1 SD ± 11.4). Additionally, differences were observed between the teachers who reported having sleep disturbances “Less than once a week” and those who reported having sleep disturbances “Once or twice a week” and “Three or more times a week” ($\tilde{x}$ = 14.4 SD ± 8.7 vs. $\tilde{x}$ = 20.4 SD ± 8.9 and $\tilde{x}$ = 30.1 SD ± 11.4).

Daytime dysfunction: Regarding daytime dysfunction, there were significant differences in emotional exhaustion between teachers who reported “Not during the past month” compared to those who reported having daytime dysfunction “Less than once a week”, ”Once or twice a week”, and “Three or more times a week” ($\tilde{x}$ = 11.8 SD ± 7.4 vs. $\tilde{x}$ = 17.9 SD ± 9.5, $\tilde{x}$ = 20.4 SD ± 9.2 and 25.5 SD ± 8.4). A significant difference in emotional exhaustion was observed between teachers reporting having daytime dysfunction “Less than once a week” and “Three or more times a week” ($\tilde{x}$ = 17.9 SD ± 9.5 vs. $\tilde{x}$ = 25.5 SD ± 8.4).

#### 3.4.2. Depersonalization

Sleep efficiency: Statistically significant differences in depersonalization were observed when comparing teachers with a sleep efficiency of less than 65 percent with respect to teachers with a sleep efficiency of 65–74 percent, 75–84 percent, and more than 85 percent ($\tilde{x}$ = 6.6 SD ± 4.3 vs. $\tilde{x}$ = 3.5 SD ± 3.7, $\tilde{x}$ = 4.2 SD ± 4.1 and $\tilde{x}$ = 4.7 SD ± 4.5).

Daytime dysfunction: Teachers who reported having daytime dysfunction “Not during the past month” were compared to those who reported having daytime dysfunction “Less than once a week” and “Once or twice a week”. Statistically significant differences in depersonalization were observed between the groups ($\tilde{x}$ = 4.0 SD ± 4.3 vs. $\tilde{x}$ = 4.9 SD ± 4.4, $\tilde{x}$ = 5.3 SD ± 4.3), respectively.

#### 3.4.3. Personal Accomplishment

Sleep quality: Significant differences in personal accomplishment were observed when comparing teachers with “Fairly bad, “Fairly good, and “Very good” sleep quality ($\tilde{x}$ = 34.1 SD ± 7.0 vs. $\tilde{x}$ = 36.7 SD ± 7.2, $\tilde{x}$ = 37.2 SD ± 7.6), respectively. Additionally, differences were observed between the teachers who reported having “Very bad” and “Fairly good” sleep quality ($\tilde{x}$ = 40.0 SD ± 6.7 vs. $\tilde{x}$ = 36.7 SD ± 7.2).

Sleep latency: When comparing the self-reported sleep onset time, teachers who reported an onset time of <15 min had significant differences in personal accomplishment measurements compared to teachers who reported an onset time of 16–30 min and 31–60 min ($\tilde{x}$ = 37.7 SD ± 7.4 vs. $\tilde{x}$ = 35.7 SD ± 7.0, $\tilde{x}$ = 35.3 SD ± 7.5).

Sleep duration: There were significant differences in measurements of personal accomplishment between the teachers who indicated they slept more than 7 h compared to those who indicated they slept less than 5 h, 5–6 h, and 6–7 h. ($\tilde{x}$ = 38.0 SD ± 7.2 vs. $\tilde{x}$ = 35.6 SD ± 6.7, $\tilde{x}$ = 36.0 SD ± 7.5 and $\tilde{x}$ = 35.9 SD ± 7.3).

Daytime dysfunction: Significant differences were found when comparing teachers who reported daytime dysfunction “Three or more times a week” vs. “Less than once a week” and “Not during the past month” ($\tilde{x}$ = 30.8 SD ± 7.6 vs. $\tilde{x}$ = 35.8 SD ± 6.5, $\tilde{x}$ = 38.7 SD ± 7.0). There were also differences between “Not during the past month”, “Less than once a week”, and “Once or twice a week” ($\tilde{x}$ = 38.7 SD ± 7.0 vs. $\tilde{x}$ = 35.8 SD ± 6.5 and $\tilde{x}$ = 33.6 SD ± 7.5).

### 3.5. Correlation between the Dimensions of Burnout Syndrome, Sleep Quality, Anxiety, and Depression

#### 3.5.1. Emotional Exhaustion, Sleep Quality, Anxiety, and Depression

Emotional exhaustion was significantly correlated with almost all variables related to sleep, except for the global sleep quality score. Emotional exhaustion was correlated with depression and anxiety (0.40 and 0.43, respectively). The values in the correlation between emotional exhaustion and sleep quality were modified by anxiety and depression (lower part of Table 5); all of them decreased, and in some cases, the difference was nonsignificant. The correlation coefficients between these scales and the sleep quality were lower; for two variables, there was no correlation (SQ and SLD), a decrease was observed in the correlation coefficients between the subscales EE and PA (−0.40 to −0.27), and the coefficients between EE and DP had no variations; additionally, there was a notable decrease in the coefficients between EE and SL (0.19 to 0.09). However, the decrease in the coefficients between EE and SL was not significant, and there was a decrease in the coefficients between EE and SD (0.32 to 0.19), both of which were significant.

#### 3.5.2. Depersonalization, Sleep Quality, Anxiety, and Depression

In the DP scale, with respect to sleep quality, no significant correlations were observed. Between the DP and PA scales, the coefficient was significant (−0.30). DP was correlated with A and D (0.16 and 0.16, respectively). When controlled for, anxiety and depression significantly increased the SLD coefficients (0.06, *p* > 0.05 to 0.10, *p* < 0.05) between the DP and PA dimensions; however, a significant decrease was observed (0.30 to 0.23).

#### 3.5.3. Personal Accomplishment, Quality of Sleep, Anxiety, and Depression

Finally, in the PA dimension of sleep disturbances, no correlations were observed between SQ, SL, SE, and USM. When the PA scale was controlled for anxiety and depression, no significant correlations were observed except for a significant decrease in DD (0.23 to 0.15).

In summary, there was a low and significant correlation between the subscales of burnout syndrome, sleep disturbance, depression, and anxiety. These correlation coefficients mostly decreased when controlling for anxiety and depression. These results suggest that depression and anxiety can affect burnout syndrome, but the relationship is not necessarily direct.

Figure 1 shows that 9.4 percent of teachers have BS, 22.5 percent have SD, while 3.0 percent live with D and 1.9 percent with A. A total of 34.4 percent of teachers are comorbid with several of these conditions (D–A, SD–A, SD–D–A, BS–D–A, BS–D, BS–SD–D–A, SD–D, BS–A, BS–SD–D, BS–SD–A and SD–BS).

The interaction between BS and SD was 5.6%, and the interaction between sleep disturbances, depression and anxiety were 7.7%. More than half of the teachers (71.2%) had emotional disorders, including burnout syndrome, sleep disturbances, depression, and anxiety. Notably, burnout syndrome can manifest itself independently of various emotional disorders. This highlights the need to establish a clear method for differential diagnosis of the different conditions impacting teachers.

### 3.6. Risk Factors for Burnout Syndrome

It is necessary to analyse and discriminate whether sleep variables along with depression and anxiety can predict burnout syndrome. In this analysis, all the variables of sleep quality, depression, and anxiety were included, and the variables that were not statistically significant were eliminated (Table 6).

#### 3.6.1. Emotional Exhaustion

The variables predicting risk of emotional exhaustion were sleep duration (β = 1.20), sleep efficiency (β = 1.15), daytime dysfunction (β = 1.18), depression (β = 1.02), and anxiety (β = 1.01). There was a significant trend in increasing emotional exhaustion generated by sleep, depression, and anxiety factors in teachers. There were other aspects that we expected to be related to emotional exhaustion, such as sleep quality (*p* < 0.09), sleep latency (*p* < 0.053), and sleep duration (*p* < 0.28); however, they were not significant.

#### 3.6.2. Depersonalization

The variables that predicted the risk of DP were sleep latency (β = 0.85), sleep duration (β = 0.84), depression (β = 1.03), and anxiety (β = 1.01). These data represent an interesting finding because these sleep variables are related to the perceptions of sleep duration and sleep onset, and we expected these values to be linked to emotional exhaustion; however, they were risk factors for depersonalization, and although the risk factor was <1, they were statistically significant. No significant differences were observed for sleep quality (*p* = 0.70), sleep efficiency (*p* = 0.35), sleep disturbances (*p* = 0.39), and daytime dysfunction (*p* = 0.60).

#### 3.6.3. Personal Accomplishment

The variables that predicted risk for personal fulfilment were daytime dysfunction (β = 1.08), depression (β = 1.0), and anxiety (β = 1.0). Personal accomplishment was strongly linked to daytime sleepiness; in other words, if a teacher is sleepy during the day, it affects their professional performance and quality of life. With respect to sleep quality (*p* = 0.50), sleep latency (*p* = 0.91), sleep duration (*p* = 0.43), sleep efficiency (*p* = 0.31), and sleep disturbances (*p* = 0.58) were factors that could be associated with emotional exhaustion, but were not statistically significant.

## 4. Discussion

In the present study, sociodemographic variables were not related to burnout, except for working more than 40 h per week and the emotional exhaustion dimension. We observed that teachers who worked more than 40 h a week or had work overload were more likely to experience emotional exhaustion. Work overload and long hours are risk factors for the development of burnout [51,52,53]. Other studies have also indicated that the number of jobs, the number of students for whom teachers are responsible, and even the sex of teachers are risk factors for the development of burnout syndrome [54]. In this study, we found no significant associations between sociodemographic variables and burnout. Therefore, our results are inconsistent with the reported literature. This may be due to the heterogeneity and inconsistency among studies on sleep disturbance, anxiety, and depression [55].

The results of this study show that the prevalence rates of burnout and poor sleep quality are high among teachers. Emotional exhaustion was significantly correlated with all sleep quality variables. There was a significant relationship between burnout and poor sleep quality in teachers. Grossi et al. [56] found that burnout syndrome was associated with poor sleep quality, more frequent awakenings, and greater sleepiness. In the present study, we found that emotional exhaustion and SB were significantly associated with all sleep quality variables.

These findings are consistent with reports that sleep quality is closely related to burnout syndrome, particularly emotional exhaustion and depersonalization [57]. In a study on the relationship between sleep quality and dimensions of burnout, significant differences were found between good and poor sleepers on the subscales of emotional exhaustion, depersonalization, and personal accomplishment [58].

We found significant differences in sleep quality parameters between participants with 65% SE and the other groups. SE is commonly defined as the relationship between total sleep time (TST) and time in bed (TIB); it plays a central role in insomnia research and practice. The struggle to fall asleep is associated with psychological distress before sleep, during the attempt to fall asleep, and during normal waking hours. This exacerbates the problem of insomnia, the prevalence of which is 20.7% in BS patients [59,60,61]. Regarding depersonalization, only sleep efficiency and daytime dysfunction led to significant differences. In addition, personal fulfilment measurements significantly differed according to SQ, SL, SLD, and DD. Some researchers have found that these indicators affect the sleep quality of people who are awake for longer than the usual 16–18 h, or who do not get enough sleep every 24 h for one or more nights. The changes in subjective sleepiness, mood, and emotional processing (including the ability to read positive emotional expressions), as well as the decreased stress threshold, lead to increased stress responses [62,63,64,65]. Moreover, they are important risk factors for the development of burnout syndrome [51,52,53]. It is well documented that sleep disturbances are associated with burnout [39,40,41], although these analyses did not report prevalence according to burnout subscales, only by the total sleep quality score.

The associations of burnout syndrome with sleep disturbance, depression, and anxiety were examined (Figure 1). These variables were independently related. The prevalence of sleep disturbances is high, affecting 22.5% of the population [66,67,68]. The rate of depression in our study population was 3.0%, which is similar to the rate reported in the general population of Mexico [6]. In other countries, the prevalence of depression is higher [7,8,9], and 6.6% of the population suffers from both sleep disturbances and depression. Chang et al. [69] found that several sleep parameters significantly differed between individuals with and without depression compared to individuals with good quality sleep. Individuals with depression were more likely to have sleep disturbances [70]. According to a longitudinal study of 4848 adults aged 50 years and older, depressive symptoms caused more sleep disturbances over a four-year period than those at baseline [71]. There was a relationship between sleep quality and depression, and it is possible that the relationship is bidirectional [72].

On the other hand, there is evidence that this bidirectional relationship does not lead to instantaneous changes; studies of other sleep disturbances such as insomnia have shown that it takes several months for people to develop depression [73,74]. However, sleep disturbances and depression are not the only related variables. Other disorders, such as anxiety and SB, are associated with sleep disturbances and depression. For example, anxiety occurred in 1.9% of teachers in this study. High prevalence rates of anxiety have been reported in the literature, ranging from 24 to 36% [9,10,68]. In this study, the prevalence of anxiety was lower because cases with comorbid depression and burnout were excluded. These percentages are similar to those previously reported in Mexico [6]. Research shows that people with poor sleep quality have high rates of depression [75,76]; anxiety disorders may be associated with different types of sleep disturbance, but anxiety symptoms are associated with poor sleep quality and difficulty maintaining sleep [68,77]. The prevalence of burnout in this study was lower than reported in the literature—in the range of 25–27% [18,78]. On the other hand, significant associations were found between burnout syndrome and sleep quality [39,41].

The sleep variables did not correlate with the depersonalization scale, but the personal accomplishment dimension was related to daytime dysfunction and sleep disturbance. Almost all sleep dimensions, except for depression and subjective sleep quality, were significantly correlated with depression and anxiety.

The correlations between sleep disturbances, depression, and anxiety were low and significant except for between sleep quality and depression. This is consistent with the reported literature [79,80,81]; other researchers observed that the correlation between de-pression and anxiety is high and significant (approximately 0.67) [82,83,84]. On the other hand, low correlations between depression and anxiety have been reported (0.29) [81].

One of the main hypotheses that guided this research was that burnout syndrome has a relationship with other conditions and overlaps with other pathologies [85,86,87]. The results in this research suggest that burnout is significantly linked with sleep disturbances, anxiety, and depression; additionally, the correlation coefficients were low and significant. In this study, emotional exhaustion was correlated with depression and anxiety (0.40 and 0.43, respectively). The correlations between depression and anxiety with depersonalization (0.16 and 0.16) and personal fulfilment (0.28 and 0.29) decreased. In general, the prevalence and correlation between burnout syndrome, depression, and anxiety found in our research support the hypothesis that these disorders are distinct entities [78,88]. However, the symptoms that occur in burnout and depression are very similar; for example, they both lead to apathy at work, and decreased productivity and vital energy of the individual [69]. These clinical features appear to be supported by the correlation observed in this study between burnout, depression, and anxiety, and are primarily associated with the dimension of emotional exhaustion. Studies conducted in other professions have reported greater correlations between the presence of depression and the scales of personal accomplishment, depersonalization, and emotional exhaustion [89,90]. According to the literature, there is a significant correlation between depression, anxiety, and the dimension of emotional exhaustion, and, to a lesser extent, with the other two dimensions [18,91]. These results support the hypothesis that burnout syndrome and, in particular, emotional exhaustion can be largely explained by depression and anxiety [86,92].

However, emotional exhaustion, depersonalization, and personal fulfilment (burnout syndrome) cannot be explained exclusively by depression and anxiety. The results in this research present a positive and significant correlation in these dimensions when con-trolled by depression and anxiety. In other words, burnout syndrome can occur independently from depression and anxiety [87,93].

In the analysis of the prevalence between burnout syndrome, sleep disturbances, depression, and anxiety (Figure 1), conditions that occurred independently were identified. It was observed that there may be cases in which the various disorders overlap due to their clinical characteristics with some frequency due to their comorbidity [94]. Taken together, it can be said that burnout syndrome is not the only disorder that overlaps with other disorders. This phenomenon is also observed, for example, between depression and anxiety. This raises the need to make a differential diagnosis between these conditions [94,95] and analyse the conceptual and methodological problem of the various diagnostic criteria of burnout in each country [96].

SLD, SE, DD, D, and A are sleep disturbances associated with burnout that predict the dimension of emotional exhaustion. SL, SLD, D, A, and DD, D, and A are sleep variables predicting personal accomplishment and depersonalization, respectively. A greater number of sleep factors influenced emotional exhaustion, while a smaller number of sleep factors could be used to predict personal fulfilment and depersonalization. It is important to note that, in the three dimensions of burnout, depression and anxiety are variables that significantly influence the prediction of this syndrome in each of the burnout scales.

### Limitations

The disadvantages of this research are related to the generalizability of the results, so it is necessary for future studies to take some of the following precautions:This study used a cross-sectional design. It is known that a limitation in this type of research is that they only assess the problem at a specific moment in time, so estimating prevalence as well as incidence requires longitudinal studies. It is also necessary to select the sample randomly to avoid sampling bias;Another disadvantage of this study was the selection of the Poisson regression analysis. The variables that were analysed in this research did not meet the normality criteria, which limited the comparison of our study with other univariate and multivariate regression analyses;A third drawback was that there are few studies that integrated burnout syndrome, sleep quality, depression, and anxiety in teachers in their analysis, which limited the ability of our analysis to elicit a better understanding of the characteristics and consequences of these phenomena on the mental health of teachers;Finally, the definition and diagnostic criteria for burnout syndrome are varied and heterogeneous, which makes it difficult to compare the results of different studies.

## 5. Conclusions

We did not observe a relationship between sociodemographic variables and burnout, except among teachers working more than 40 h and the emotional exhaustion dimension. The findings in this study indicate that the prevalence of burnout and poor sleep quality in the teaching population is high. There is a significant correlation between the emotional exhaustion dimension and all sleep quality variables. The depersonalization scale was not correlated with sleep variables, whereas the personal accomplishment dimension was associated with sleep disturbance and daytime dysfunction.

Depression and anxiety have significant correlations with the burnout dimensions and with almost all sleep dimensions, except for depression with subjective sleep quality. These correlation coefficients decreased markedly after controlling for depression and anxiety variables.

The symptoms of burnout syndrome in teachers may overlap with sleep disturbances. By eliminating the influence of depression and anxiety, these disorders can be distinguished; however, a differential diagnosis is needed due to the comorbidity present in these conditions, as they can be considered independent disorders.

## Figures and Tables

**Figure 1 ijerph-20-06276-f001:**
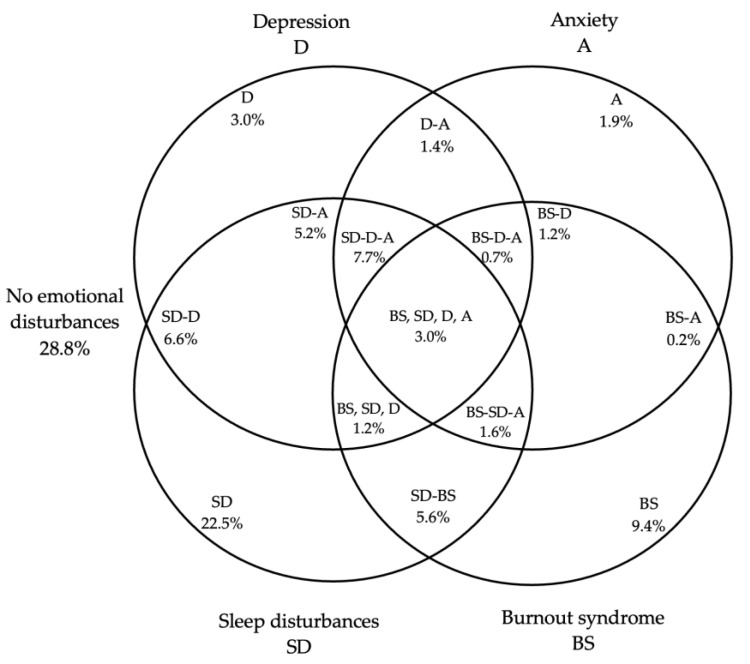
Interaction between burnout syndrome, sleep disturbances, depression, and anxiety.

**Table 1 ijerph-20-06276-t001:** Sociodemographic and sleep characteristics of teachers.

Variable		N	%	SQ	SL	SLD	SE	SD	USM	DD	* *p* < 0.05
Sex											
	*Female*	256	60.0	214.06	211.20	212.56	214.02	217.21	215.18	215.70	NS
	*Male*	171	40.0	213.91	218.19	216.15	213.96	209.19	212.23	211.45	
Age											
	*22–35 (a)*	128	30.0	133.00	134.75	130.31	131.81	134.06	133.38	126.31 *	* c
	*36–46 (b)*	155	36.3	140.87 *	146.90	144.05	148.68	146.19	149.07	139.61	* c
	*47–64 (c)*	144	33.7	159.83	153.34	156.41	151.42	154.10	151.00	161.18	
Marital status											
	*Married (a)*	281	65.8	219.32	208.79	208.48	212.11	211.19	211.42	211.55	
	*Single (b)*	87	20.4	207.67	213.63	223.30	213.45	219.35	208.32 *	210.72	* c
	*Other (c)*	59	13.8	198.02	239.37 *	226.58	223.81	219.52	234.69 *	230.52	* a
Children											
	*0 (a)*	82	19.2	132.13	123.57	125.72	135.07	135.80	133.24	133.39	NS
	*1–2 (b)*	238	55.7	121.99	125.55	120.99	120.45	121.55	126.13	124.55	NS
	*3+ (c)*	107	25.1	128.78	124.35	132.70	130.93	128.54	120.28	123.35	NS
Educational Level											
	*Primary (a)*	251	58.8	214.75	225.69 *	222.66 *	215.47	223.50 *	216.36	216.63	b *
	*Secondary (b)*	176	41.2	212.92	197.33	201.64	211.90	200.46	210.63	210.24	
Working hours per week											
	*5–15 (a)*	23	5.4	132.13	123.57	125.72	135.07	135.80	133.24	133.39	NS
	*16–27 (b)*	150	35.1	121.99	125.55	120.99	120.45	121.55	126.13	124.55	NS
	*28–39 (c)*	76	17.8	128.78	124.35	132.70	130.93	128.54	120.28	123.35	NS
	*40+ (d)*	178	41.7	132.13	123.57	125.72	135.07	135.80	133.24	133.39	NS
Number of students per group											
	*10–20 (a)*	28	6.6	243.25	220.38	224.00	222.13	211.38	206.13	222.38	
	*21–30 (b)*	91	21.3	220.96	237.38 *	224.00	221.54	211.96	231.35 *	218.27	* d
	*31–40 (c)*	110	25.8	216.08	201.94 *	214.02	213.11	206.52	206.26 *	213.64	* b
	*41+ (d)*	198	46.4	205.51	209.05	207.98	209.88	219.46	211.44	211.05	
Number of education jobs											
	*1*	201	47.1	219.35	215.86	216.41	215.87	218.54	214.43	221.05	NS
	*2*	226	52.9	209.24	212.35	211.85	212.34	209.96	213.62	207.73	
Type of employment											
	*Permanent (a)*	393	92.0	213.64	214.26	215.62	213.88	211.78	214.80	213.55	NS
	*Eventual (b)*	15	3.5	207.67	195.47	217.90	240.93	229.17	212.73	211.70	NS
	*Both (c)*	19	4.5	226.39	223.18	177.45	195.24	247.89	198.50	225.18	NS

SQ (Sleep Quality), SL (Sleep Latency), SLD (Sleep Duration), SE (Sleep Efficiency), SD (Sleep Disturbances), USM (Use of Sleep Medication), DD (Daytime Dysfunction). The sleep variables are presented in their average ranges. NS = No significant.

**Table 2 ijerph-20-06276-t002:** Relationship between sociodemographic variables and burnout syndrome dimensions.

	Emotional Exhaustion		Depersonalization		Personal Accomplishment	
Variable	Mean	SD	Mean	SD	Mean	SD
Sex						
*Female*	16.1	9.4	4.2	4.3	36.4	7.3
*Male*	15.9	9.5	5.1 *	4.5	36.4	7.3
Age						
*22–35*	16.1	9.2	4.4	4.4	35.4	7.3
*36–46*	16.1	9.8	5.0	4.7	36.3	6.7
*47–64*	16.0	9.4	4.3	4.0	37.5 *	7.8
Marital status						
*Married*	15.8	9.6	4.4	4.3	36.5	7.1
*Single*	16.01	9.4	4.9	4.3	36.5	7.7
*Other*	17.3	8.8	5.0	4.8	36.0	7.7
Children						
*0*	17.2	9.5	4.9	4.3	35.8	7.8
*1–2*	15.5	9.0	4.2	4.4	36.4	7.2
*3+*	16.4	10.2	5.3 *	4.3	37.0	7.2
Educational level						
*Primary*	16.6	9.4	4.3	4.0	36.4	7.6
*Secondary*	15.3	9.4	4.8	5.0	36.4	6.8
Working hours per week						
*5–15*	11.9	5.4	6.1	5.7	7.0	1.5
*16–27*	15.0	9.4	4.2	4.1	36.8	7.0
*28–39*	15.9	10.4	5.2	4.9	36.1	7.2
*40+*	17.5 *	9.3	4.5	4.1	36.2	7.8
Number of students per group						
10–20	14.5	9.3	4.3	3.6	37.1	7.5
21–30	15.4	8.2	4.1	4.1	36.0	7.6
31–40	15.1	9.8	4.9	4.3	35.2	7.4
41+	16.8	9.8	4.7	4.7	36.8	7.1
Number of education jobs						
*1*	16.3	9.5	4.7	4.6	36.3	7.5
*2*	15.8	9.5	4.5	4.1	36.5	7.2
Type of employment						
*Permanent*	16.0	9.4	4.6	4.4	36.5	7.1
*Eventual*	15.9	10.9	6.5	4.6	36.6	8.9
*both*	17.7	9.1	3.5	3.5	35.5	9.6

Standard deviation (SD). * *p* < 0.05.

**Table 3 ijerph-20-06276-t003:** Features of the PSQI.

Variable		N	%
SQ	*Very good*	22	5.2
	*Fairly good*	250	58.5
	*Fairly bad*	106	24.8
	*Very bad*	49	11.5
SL	<15 min: *Not during the past month*	191	44.7
	16–30 min: *Less than once a week*	142	33.3
	*31–60 min: Once or twice a week*	74	17.3
	*>60 min: Three or more times a week*	20	4.7
SLD	*>7 h*	106	24.8
	*6–7 h*	158	37.0
	*5–6 h*	111	26.0
	*<5*	52	12.2
SE	* ≥85%*	273	63.9
	*75–84%*	94	22
	*65–74%*	38	8.9
	*<65%*	22	5.2
SD	*Not during the past month*	20	4.7
	*Less than once a week*	295	69.1
	*Once or twice a week*	104	24.4
	*Three or more times a week*	8	1.9
USM	*Not during the past month*	396	92.7
	*Less than once a week*	19	4.4
	*Once or twice a week*	6	1.4
	*Three or more times a week*	6	1.4
DD	*Not during the past month: No problem at all*	182	42.6
	*Less than once a week: Only a very slight problem*	155	36.3
	*Once or twice a week: Somewhat of a problem*	71	16.6
	*Three or more times a week: A very big problem*	19	4.4

SQ (Sleep Quality), SL (Sleep Latency), SLD (Sleep Duration), SE (Sleep Efficiency), SD (Sleep Disturbances), USM (Use of Sleep Medication), DD (Daytime Dysfunction).

**Table 4 ijerph-20-06276-t004:** Burnout syndrome and PSQI.

	Emotional Exhaustion		Depersonalization		Personal Accomplishment	
Variable	Mean	SD	Mean	SD	Mean	SD
SQ						
*Very good (a)*	19.9 *d* *	11.6	5.8	5.3	37.2 *c* *	7.6
*Fairly good (b)*	14.9 *c* *, *d* **	8.9	4.6	4.4	36.7 *c* *	7.2
*Fairly bad (c)*	20.3 *d* *	8.9	4.9	4.4	34.1	7.0
*Very bad (d)*	10.9	7.9	3.6	3.8	40.0 *b* *	6.7
SL						
<15 min: *Not during the past month (a)*	14.2	9.2	4.7	4.2	37.5	7.4
16–30 min: *Less than once a week (b)*	16.4 *a* *, *d* **	9.3	4.6	4.6	35.7 *a* **	7.0
*31–60 min: Once or twice a week (c)*	18.5 *a* *	9.5	4.2	4.6	35.3 *a* **	7.5
*>60 min: Three or more times a week (d)*	22.5 *a* *	8.0	4.9	3.4	35.5 *a* *	6.9
SLD						
*>7 h (a)*	13.4	8.0	4.8	4.6	38.0	7.2
*6–7 h (b)*	15.6 *d* **	8.9	4.8	4.3	35.9 *a* **	7.3
*5–6 h (c)*	17.7 *a* **	10.7	4.3	4.2	36.0 *a* **	7.5
*<5 h (d)*	19.2 *a* *	9.7	4.2	4.5	35.6 *a* **	6.7
SE						
*>85% (a)*	15.0	9.0	4.7 *d* **	4.5	36.8	7.1
*75–84% (b)*	16.0 *c***	9.2	4.2 *d* **	4.1	36.3	7.6
*65–74% (c)*	20.5 a *	9.4	3.5 *d* *	3.7	34.8	7.7
*<65% (d)*	21.6 a **	11.9	6.6	4.3	34.8	7.5
SD						
*Not during the past month (a)*	11.5	9.8	4.0	2.8	39.4	7.1
*Less than once a week (b)*	14.4 a *	8.7	4.6	4.4	36.7	7.3
*Once or twice a week (c)*	20.4 b *	8.9	4.6	4.2	35.2	7.2
*Three or more times a week (d)*	30.1 a *, b *	11.4	6.4	7.3	34.3	7.5
USM						
*Not during the past month (a)*	15.9	9.6	4.6	4.4	36.5	7.3
*Less than once a week (b)*	18.0	7.5	4.0	4.3	36.0	6.8
*Once or twice a week (c)*	19.8	6.9	3.5	2.7	34.3	8.3
*Three or more times a week (d)*	18.0	9.0	4.2	4.2	35.8	9.7
DD						
*Not during the past month: No problem at all (a)*	11.8	7.4	4.0	4.3	38.7 d *	7.0
*Less than once a week: Only a very slight problem (b)*	17.9 a *,	9.5	4.9 a **	4.4	35.8 a *, d **	6.5
*Once or twice a week: Somewhat of a problem (c)*	20.4 a *	9.2	5.3 a **	4.3	33.6 a *	7.5
*Three or more times a week: A very big problem (d)*	25.5 a *, b **	8.4	4.8	4.5	30.8	7.6

The variables in each category are represented by lower case letters, which are used to compare and identify statistically significant differences. *****
*p* < 0.01 ******
*p* < 0.05. Standard deviation (SD). Confidence Interval (CI).

**Table 5 ijerph-20-06276-t005:** Correlation between burnout syndrome, sleep quality, anxiety, and depression.

	1	2	3	4	5	6	7	8	9	10	11	12
1. EE	1	0.27 **	−0.40 **	0.10 *	0.19 **	0.16 **	0.20 **	0.32 **	0.09	0.30 **	0.40 **	0.43 **
2. DP		1	−0.30 **	−0.01	−0.04	−0.06	0.01	0.02	−0.04	0.08	0.16 **	0.16 **
3. PA			1	−0.05	−0.08	−0.06	−0.09	−.102*	−0.03	−0.23 **	−0.28 **	−0.29 **
4. SQ				1	0.15 **	0.20 **	0.13 **	0.12 **	0.09	0.24 **	0.10	0.18 **
5. SL					1	0.16 **	0.24 **	0.22 **	0.33 **	0.20 **	0.20 **	0.26 **
6. SLD						1	0.40 **	0.20 **	0.12 *	0.22 **	0.15 **	0.22 **
7. SE							1	0.17 **	0.15 **	0.11*	0.16 **	0.22 **
8. SD								1	0.10 *	0.24 **	0.24 **	0.37 **
9. USM									1	0.17 **	0.12 *	0.19 **
10 DD										1	0.25 **	0.39 **
11. D											1	0.67 **
12. A												1
1. EE	1	0.27 **	−0.27 **	0.02	0.09	0.10 *	0.13 **	0.19 **	−0.01	0.16 **		
2. DP		1	−0.23 **	−0.06	−0.08	−0.10 *	−0.03	−0.04	−0.07	0.01		
3. PA			1	0.00	0.00	0.00	−0.03	0.01	0.02	−0.15 **		
4. SQ				1	0.11 *	0.16 **	0.09	0.07	0.06	0.19 **		
5. SL					1	0.12 *	0.20 **	0.15 **	0.30 **	0.12 *		
6. SLD						1	0.38 **	0.13 *	0.08	0.15 **		
7. SE							1	0.10 *	0.11 **	0.03		
8. SD								1	0.040	0.12 *		
9. USM									1	0.11 *		
10. DD										1		

Note: ** *p* < 0.01 level; * *p* < 0.05 level. EE (Emotional Exhaustion), DP (Depersonalization), PA (Personal Accomplishment), D and A (Depression and Anxiety). SQ (Sleep Quality), SL (Sleep Latency), SLD (Sleep Duration), SE (Sleep Efficiency), SD (Sleep Disturbances), USM (Use of Sleep Medication), DD (Daytime Dysfunction).

**Table 6 ijerph-20-06276-t006:** Predictor values for burnout syndrome dimensions.

**Emotional Exhaustion**			**95% Wald** **Confidence Interval**						**95% Wald** **Confidence Interval**	
**Parameter**	**B**	**Std. Error**	**Lower**	**Upper**	**Wald Chi-Square**	**df**	**Sig.**	**Exp (B)**	**Lower**	**Upper**
Intercept	2.37	0.02	2.33	2.41	11,440.98	1	0.00	10.71	10.25	11.19
SLD	0.18	0.03	0.13	0.24	43.50	1	0.00	1.20	1.14	1.27
SE	0.14	0.03	0.07	0.20	17.66	1	0.00	1.15	1.08	1.22
DD	0.17	0.03	0.11	0.22	32.63	1	0.00	1.18	1.12	1.25
D	0.02	0.00	0.02	0.03	59.54	1	0.00	1.02	1.02	1.03
A	0.01	0.00	0.01	0.02	46.68	1	0.00	1.01	1.01	1.02
**Depersonalization**			**95% Wald** **Confidence Interval**						**95% Wald** **Confidence Interval**	
**Parameter**	**B**	**Std. Error**	**Lower**	**Upper**	**Wald Chi-Square**	**df**	**Sig.**	**Exp (B)**	**Lower**	**Upper**
Intercept	1.32	0.04	1.23	1.40	997.40	1	0.00	3.73	3.44	4.05
SL	−0.16	0.06	−0.27	−0.05	7.97	1	0.00	0.85	0.76	0.95
SLD	−0.17	0.05	−0.27	−0.07	12.17	1	0.00	0.84	0.77	0.93
D	0.03	0.01	0.02	0.04	29.74	1	0.00	1.03	1.02	1.04
A	0.01	0.00	0.00	0.02	9.33	1	0.00	1.01	1.00	1.02
**Personal Accomplishment**			**95% Wald** **Confidence Interval**						**95% Wald** **Confidence Interval**	
**Parameter**	**B**	**Std. Error**	**Lower**	**Upper**	**Wald Chi-Square**	**df**	**Sig.**	**Exp (B)**	**Lower**	**Upper**
Intercept	3.61	0.03	3.56	3.66	20,340.10	1	0.00	36.9	35.1	38.7
DD	0.08	0.02	0.04	0.12	13.01	1	0.00	1.08	1.04	1.13
D	−0.01	0.00	−0.01	0.00	11.10	1	0.00	1.00	0.99	1.00
A	0.00	0.00	−0.01	0.00	6.76	1	0.01	1.00	0.99	1.00

## Data Availability

Not applicable.

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
