# Peer review of "Burnout Syndrome and Sleep Quality in Basic Education Teachers in Mexico"

_ijerph, 2023, doi:10.3390/ijerph20136276_

Round 1

Reviewer 1 Report

There are some concerns:

1.In the introduction, 1.1, authors describe "burnout syndrome, depression and anxiety", but it is still not much clear about the criteria, the similarities and the differences between burnout syndrome and depression/anxiety. Could authors summarize more about these?

2.In the results, emotional exhaustion affects most sleep parameters, SQ, SL, SLD,SE, SD and DD, while depersonalization has a weaker effect, are there some explanations?

3.And emotional exhaustion is only significant in working hours >40 hours, could it explain why teachers working <40 hours have poor sleep?

4. Could authors add another table at the basis of table 2, showing the sleep quality of each variation in sex, age, working hours... This would make it more straightforward to tell the effect of variable conditions, EE/DD/PA, on sleep quality.

Author Response

Dear reviewer

We have sent the responses to the recommendations and include a certificate of editing of the manuscript at the end of the responses to address the point "Extensive revision of English language and style required" and we will revise the editing of the manuscript as often as necessary.

We hope that we have considered and implemented the recommendations in the manuscript.

Kind regards

The authors

Reviewer 2 Report

There is an issue with the term “sleep disorders” as the authors included sleep disturbances (assessed by the Pittsburgh Sleep Quality Index questionnaire) as a sleep disorder. Examples of sleep disorders include insomnia disorder, narcolepsy, restless leg syndrome, sleep apnea etc. but this manuscript does not address any of these disorders.  The anxiety and depressive disorders are not classified under sleep disorders. Therefore, please replace "sleep disorders" with either "sleep disturbances" or "poor sleep quality" throughout the manuscript since you did not assess sleep disorders. Because of this, we suggest that the authors review the entire manuscript for any discrepancies. And please also revise the research title.

The Introduction is well written.

Statistical analysis: It will be helpful to provide more details about the statistical tests used in the study (to which set of data applied).

Section 2.4: Please specify if mediation testing was applied (see Line 367), and if so, explain why regression analysis was not applied. If not, please replace “mediate” with a more appropriate term/word.

Results:  A lot of descriptive data were repeated, i.e., same data description on the text and in Tables. Please reduce repetitions by highlight new information on the text, e.g., Lines 243-254 and contents of Table 1.

Discussions: Please move the first two paragraphs of the “Conclusion” to the start of the Discussion section. In this way, the findings of the study are first reiterated to prompt discussions, where you have two major topics for discussions: 1) significant correlation between the emotional exhaustion dimension and all sleep quality variables; and 2) Depression and anxiety have significant correlations with the burnout dimensions and with almost all sleep dimensions, except depression with subjective sleep quality. For this discussion, the authors are recommended to address the interaction between sleep disturbances and depression (a bi-directional relationship) and between sleep disturbances and anxiety, and how these contribute to the burnout syndrome.

It is suggested that the section in Line 480-486 should be re-written in a Discussion style rather than presenting it in a Result style. 

There is no evidence to suggest that sleep quality items were related to sleep deprivation (Line 461-462) or fragmentation. Poor sleep quality does not need to be equated to sleep deprivation.

Specific comments:

Line 24: please re-write this phrase.

Line 160, 167: please replace “application” with “administration”

Line 164: please re-write this phrase - please clarify what it means by "at the school supervision and are determined by the supervisor"

Line 173: Do you mean "ethical approval"? Please re-write this phrase.

Line 181: Do you mean "the scores for each dimension were categorisezed ..."?

Line 195: The measure consists of 19 individual items, creating 7 components that produce one global score. Where did the 24 questions come from? Please clarify.

Line 197: Please delete "that are" and replaced them with "with each scored on a scale from 0 - 3".

Line 198: please delete "higher score indicates lower" and replace it with "a score greater than 5 indicates poor sleep quality".

Line 273: No brackets required for Table 2.

Line 277: Please clarify what was "the expected range".

Tables: SD = sleep disturbance, please select a different acronym so to not confuse with SD (standard deviation).

Line 289, Table 4 brackets not required.

Line 345: Please replace "sleep disorders" with either "PSQI items" or "sleep quality"

Line 365: Since you are referring to the PSQI global sleep quality score. it will be helpful to say "except for the global sleep quality score".

Line 488: if depression and anxiety are distinct entities, how did they impact the findings? Please clarify.

Line 517: please change "disorder" to "syndrome" (Burnout is a syndrome, not a disorder). 

Author Response

Dear reviewer

We have sent the responses to the recommendations and include a certificate of editing of the manuscript at the end of the responses to address the point " Moderate English changes required " and we will revise the editing of the manuscript as often as necessary.

We hope that we have considered and implemented the recommendations in the manuscript.

Kind regards

The authors
